# Prevalence, Management, and Associated Factors of Obesity, Hypertension, and Diabetes in Tibetan Population Compared with China Overall

**DOI:** 10.3390/ijerph19148787

**Published:** 2022-07-19

**Authors:** Wen Peng, Ke Li, Alice F. Yan, Zumin Shi, Junyi Zhang, Lawrence J. Cheskin, Ahktar Hussain, Youfa Wang

**Affiliations:** 1Nutrition and Health Promotion Center, Department of Public Health, Medical College, Qinghai University, Xining 810016, China; wen.peng2014@foxmail.com; 2Global Health Institute, School of Public Health, Xi’an Jiaotong University, Xi’an 710049, China; likekkzk@163.com; 3Division of Research Patient Care Services, Stanford Health Care, Palo Alto, CA 94305, USA; dralicey@gmail.com; 4Human Nutrition Department, College of Health Sciences, QU Health, Qatar University, Doha 2713, Qatar; zumin@qu.edu.qa; 5Department of Clinical Nutrition, The Third People’s Hospital of Chengdu, Chengdu 610014, China; zhangjunyi9@hotmail.com; 6Department of Nutrition and Food Studies, College of Health and Human Services, George Mason University, Fairfax, VA 22030, USA; lcheskin@gmu.edu; 7Johns Hopkins School of Medicine, Baltimore, MD 21218, USA; 8International Diabetes Federation (IDF), 1000 Brussels, Belgium; hussain.akhtar@nord.no; 9Faculty of Health Sciences, Belgian and Nord University, 8001 Bodø, Norway

**Keywords:** obesity, diabetes, hypertension, Tibetan, prevalence, control

## Abstract

Tibetans’ life expectancy lags behind China’s average. Obesity and noncommunicable diseases (NCDs) contribute to health disparity, but NCD patterns among Tibetans are unknown. To examine the prevalence, management, and associated factors for obesity, hypertension, and diabetes among Tibetans, compared with China’s average, we systematically searched PubMed and China National Knowledge Infrastructure databases for studies between January 2010 and April 2021. Thirty-nine studies were included for systematic review, among thirty-seven that qualified for meta-analysis, with 115,403 participants. Pooled prevalence was 47.9% (95% CI 38.0–57.8) for overweight/obesity among adults (BMI ≥ 24 kg/m^2^) and 15.4% (13.7–17.2) among children using Chinese criteria, which are lower than the national rates of 51.2% and 19.0%, respectively. The estimate for hypertension (31.4% [27.1–35.7]) exceeded China’s average (27.5%), while diabetes (7.5% [5.2–9.8]) was lower than average (11.9%). Men had a higher prevalence of the three conditions than women. Residents in urban areas, rural areas, and Buddhist institutes had monotonically decreased prevalence in hypertension and diabetes. Awareness, treatment, and control rates for hypertension and diabetes were lower than China’s average. Urban residence and high altitude were consistent risk factors for hypertension. Limited studies investigated factors for diabetes, yet none exist for obesity. Tibetans have high burdens of obesity and hypertension. Representative and longitudinal studies are needed for tailored interventions. There are considerable variations in study design, study sample selection, and data-analysis methods, as well as estimates of reviewed studies.

## 1. Introduction

Noncommunicable diseases (NCDs) are the leading cause of death worldwide and in China—representing 71% and 89% of all deaths, respectively [1,2]. Meanwhile, NCDs may facilitate the health burden of communicable diseases. For example, people with NCDs have experienced disproportionately higher rates of COVID-19 hospitalization and mortality. Minority or indigenous populations are particularly vulnerable to obesity and related NCDs in some developed countries [3,4]. Data are scarce in developing countries, including China.

The Tibetan population are a unique indigenous population, who traditionally live on the highest plateau in the world—the Qinghai–Tibet Plateau—with an average altitude of more than 4500 m above sea level. Regarding the Tibetan population, around 6 million are living in China, while some live in neighboring countries, notably India because of some historical reasons. In China, they mainly live in five provincial-level administrative areas—the Tibetan Autonomous Region (TAR) and Qinghai, Sichuan, Yunnan, and Gansu provinces. The TAR has the largest Tibetan population, 3.14 million, accounting for 86% of the total population in the TAR. Thus, some indicators for the TAR can be the proxy for the Tibetan population. In this article, we use the term “Tibetan” to describe the population of Tibetan ethnicity, “TAR” to describe the provincial-level administrative areas in China, and “Tibetan areas” to describe the areas (mainly the five provinces in Western China) where Tibetans live.

The unique hypobaric hypoxia environment and other related factors may have caused the suboptimal health status among the Tibetan population, to which NCDs have contributed largely. Their poor health status was indicated by a large gap in the average life expectancy between the Tibetan and benchmark populations. The average estimate for Tibetans in 2019 was 70.6 years, far below China’s national average of 76.8 years and that of developed areas in China such as Shanghai (83.7 years).

The health burden due to overall NCDs was also high among Tibetans. NCD-attributed standardized mortality rates (SMRs) in the TAR were among the highest category of 719–867/100,000 in China. To be specific, the SMRs in the TAR were in the highest category in cardiovascular diseases (387.1–460.8/100,000), but were the second-lowest category in diabetes (4.0–7.4/100,000). However, the mortality rates in cardiovascular diseases and diabetes in China in 2018 were 364.6 and 19.1 per 100,000, respectively, based on a recent national report [5]. In addition, recent epidemiological studies in different settings have revealed an unexpectedly high prevalence of obesity and related NCDs [6,7,8]. Nevertheless, the obesity and NCD patterns among the overall Tibetan populations were understudied. Further, the variations by the unique environmental and social factors, such as altitude, gender, and religious beliefs, were not yet comprehensively investigated.

To address these research gaps, this study selected obesity and two major NCDs—hypertension and diabetes—as targeted health outcomes and indicators for NCD disparities. This study aimed to (1) systematically examine the prevalence of obesity, hypertension, and diabetes among Tibetans and by gender and age groups; (2) examine the awareness, treatment, and control rates of hypertension and diabetes among Tibetans; (3) compare the health burden of targeted health outcomes in Tibetans with China’s national average; and (4) identify associated risk factors for targeted health outcomes in Tibetans. Findings can yield valuable insights to guide future research and public-health-policy development for similar indigenous populations in China and other countries.

## 2. Materials and Methods

### 2.1. Data Sources and Search Strategies

Guided by the Preferred Reporting Items for Systematic Reviews and Meta-Analyses (PRISMA) guidelines, a systematic search was performed in two academic databases: PubMed and China National Knowledge Infrastructure (CNKI). The following search terms were used: (“Tibet*”) AND (“obes*” OR “overweight*” OR “body mass index” OR “BMI” OR “adipos*” OR “waist circumference” OR “diabetes” OR “prediabetes” OR “impaired glucose tolerance” OR “impaired fasting glycemia” OR “hypertension” OR “high blood pressure”). The references of enrolled publications were also screened for eligibility. The PRISMA flowchart of study selection is shown in Figure 1. The study protocol was registered with PROSPERO, number CRD42021255694.

### 2.2. Study Selection: Inclusion and Exclusion Criteria

Studies were included if they: (a) were cross-sectional or longitudinal studies published in peer-reviewed journals; (b) were written in English or Chinese; (c) reported prevalence of overweight/obesity and/or central obesity and/or hypertension and/or diabetes and/or pre-diabetes; (d) reported results for the Tibetan population; (e) had samples that did not overlap with other identified studies (if more than one publication used the same data, only the publication with the largest sample size was included); and (f) were published between January 2010 and April 2021.

Studies were excluded if they: (a) were qualitative studies, case reports, editorials, reviews, or meta-analyses; (b) did not provide data on any of the outcomes of interest; (c) surveyed before 2010; and (d) studied special populations, such as pregnant women, college students, etc.

### 2.3. Diagnostic Criteria for Overweight/Obesity, Hypertension, and Diabetes

Among adults, overweight/obesity was defined as a body mass index (BMI) ≥ 24 kg/m^2^ based on the Chinese criteria [9], or a BMI ≥ 25 kg/m^2^ based on the World Health Organization (WHO) criteria [10]. Central obesity was defined as waist circumference (WC) ≥ 90 cm for men and WC ≥ 80 cm for women based on the International Diabetes Federation (IDF) criteria [11]. Among children, it was defined according to the Chinese sex–age-specific BMI cutoff points [12].

Hypertension was defined as a systolic and/or diastolic blood pressure ≥ 140/90 mmHg, and/or a history of hypertension, and/or reported current treatment with antihypertensive medications [13]. Diabetes was defined as self-reported diabetes, and/or fasting plasma glucose ≥ 7.0 mmol/L, and/or two-hour plasma glucose ≥ 11.1 mmol/L; pre-diabetes was defined as any participants without diabetes but with a fasting plasma-glucose level of 5.6 mmol/L to 6.9 mmol/L, and/or two-hour plasma-glucose level of 7.8 mmol/L to 11.0 mmol/L [14]. 

### 2.4. Data Extraction and Quality Assessment

Two reviewers (K.L. and W.P.) independently extracted the following data using a standardized study form: (a) study-level characteristics (e.g., country/area, study setting, sample size, altitude, author, and survey year); (b) sample characteristics (age, gender, residence, and livelihood); (c) prevalence of overweight/obesity, central obesity, hypertension, diabetes, and pre-diabetes; (d) awareness, treatment, and control rates of hypertension and diabetes; and (f) effect sizes (e.g., Pearson’s correlation coefficient, β coefficient, or odds ratio (OR) for targeted outcomes). Disagreements were discussed to resolve the inconsistencies.

Prevalence estimates of China’s national average were from nationally representative studies conducted by the Chinese Center of Disease Control and Prevention [2,15,16,17]. For comparison, estimates were extracted for the United States (US) and the globe. Estimates in the US were derived from the National Health and Nutrition Examination Survey [18,19]. Global-prevalence estimates were from reports by the WHO or IDF [10,20,21].

The same two reviewers independently assessed the quality of the articles using the US National Institute of Health Quality Assessment Tool for Observational Cohort and Cross-Sectional Studies [22]. Overall quality was rated based on the total score of the 14 items: ≥7 was rated as good, 4–7 as fair, and <4 as poor.

### 2.5. Statistical Analysis

Prevalence estimates were calculated by pooling the study-specific estimates using a random-effect (the DerSimonian–Laird method) meta-analysis that accounted for between-study heterogeneity. Subgroup meta-analyses were conducted when stratified results were reported, such as by gender, age group, and urban/rural residence. Possible publication bias was identified by Egger’s test. Sensitivity analyses were conducted by leave-one-out method to show the effect of individual studies. Pooled prevalence estimates among Tibetans were compared with overall estimates for China, the US, and the globe.

Study heterogeneity was assessed using the I^2^ index. The level of heterogeneity was interpreted as modest (I^2^ ≤ 25%), moderate (25% < I^2^ ≤ 50%), substantial (50% < I^2^ ≤ 75%), or considerable (I^2^ > 75%). All statistical analyses were conducted in STATA with specific commands after meta set (e.g., meta summarize, meta forest-plot) (Version 17.0; Stata Corp., College Station, TX, USA). All analyses used two-sided tests, and *p*-value < 0.05 was considered statistically significant.

## 3. Results

### 3.1. Characteristics of Studies Included

Of the 448 studies screened (200 in English, 248 in Chinese), 101 were identified as potentially relevant articles, and those full-text papers were independently reviewed. In total, 39 studies (17 in English, 22 in Chinese) met the inclusion criteria for systematic review [23,24,25,26,27,28,29,30,31,32,33,34,35,36,37,38,39,40,41,42,43,44,45,46,47,48,49,50,51,52,53,54,55,56,57,58,59,60,61], of which 37 studies with consistent diagnostic criteria (115,403 participants) were included in the meta-analyses (Figure 1) [25,26,27,28,29,30,31,32,33,34,35,36,37,38,39,40,41,42,43,44,45,46,47,48,49,50,51,52,53,54,55,56,57,58,59,60,61].

Data for overweight/obesity, hypertension, and diabetes were pooled separately in the meta-analysis. Appendix A shows study characteristics, key findings, and quality ratings of the 39 studies. All were cross-sectional studies; none were longitudinal studies. Two studies used nationally representative survey data. Seven studies were rated as good quality, twenty-six as fair, and six as poor. Appendix A shows the study quality rating of the 39 studies. No publication bias was identified for overweight/obesity and diabetes studies, but publication bias did exist for hypertension studies by Egger’s test (Appendix A). Sensitivity analysis using the leave-one-out method for hypertension prevalence did [26,27] not show large impacts by any individual study (Appendix A).

Among the 37 studies included in the meta-analysis, 28 were on hypertension, 19 on overweight/obesity or central obesity, and 14 on diabetes or pre-diabetes. Regarding participants’ age distribution, 32 studies were conducted among adults, and 5 studies were among children. The studies included were geographically diverse: 36 studies were conducted in the five provinces in Western China, the so-called Tibetan areas in China; and 1 study included subjects from both China and India. Specifically, 20 studies were performed in the TAR, 7 studies in Sichuan province, 5 in Gansu province, 4 in Qinghai province, and 2 in Yunnan province.

### 3.2. Pooled-Prevalence Estimates of Overweight/Obesity, Hypertension, and Diabetes among Tibetans, and Comparison with Average Estimates for China, the US, and the Globe

The prevalence of overweight/obesity, hypertension, and diabetes varied substantially among the included studies. In meta-analyses, most of the I^2^ were above 90%. The pooled-prevalence estimates for overweight/obesity among adults, based on the Chinese (BMI ≥ 24 kg/m^2^) and WHO (BMI ≥ 25 kg/m^2^) criteria, were 47.9% (95% CI 38.0–57.8) and 31.6% (95% CI 20.9–42.3), respectively, while the estimate for central obesity was 43.5% (95% CI 40.4–46.6); the pooled estimate among children was 15.4% (95% CI 13.7–17.2) (Figure 2). Stratified analyses showed that the overweight/obesity prevalence estimate for men was higher than for women (57.7% vs. 48.2%), while central obesity showed the opposite pattern (Appendix A).

The prevalence of hypertension in adults varied from 8.4% to 64.6% in the 25 studies. For the 13 studies reporting diabetes prevalence, the rates varied from 0.9% to 27.0%. The pooled-prevalence estimates for hypertension and diabetes were 31.4% (27.1–35.7) and 7.5% (5.2–9.8), respectively, while the estimate for pre-diabetes was 20.2% (11.5–28.8) (Figure 3). Stratified analyses showed a higher pooled-prevalence estimate among men than women for both hypertension and diabetes, and a monotonic decrease in estimates among residents in urban areas, rural areas, and Buddhist institutes for both conditions (Appendix A).

Compared with average estimates for China, the US, and the globe, the pooled estimates of overweight/obesity among Tibetan adults were lower using the WHO criteria (Figure 4A). When Chinese criteria were applied, China’s national average estimates among adults in 2010, 2013, 2015, and 2018 were 42.7%, 46.9%, 47.7%, and 51.2%, respectively. Estimates among children (7–17 years) in 2002, 2012, and 2017 were 6.5%, 16.0%, and 19.0%, respectively, compared to the pooled estimate of 15.4% (13.7–17.2), using the Chinese standard. We did not find data on overweight/obesity among Tibetan children using the WHO criteria. The pooled estimate of hypertension in Tibetan adults was higher than the recent estimates for China and the globe (hypertension 31.4%, 27.5%, and 22.0%, respectively) and that of diabetes was lower than the recent estimates for China, the US, and the globe (7.5%, 11.9%, 14.6%, and 9.3%, respectively) (Figure 4) [2,10,15,16,17,18,19,20,21].

Diagnostic criteria: (1) overweight/obesity among adults was defined as BMI ≥ 25 kg/m^2^ (WHO criteria); (2) hypertension among adults was defined as systolic and/or diastolic blood pressure ≥ 140/90 mmHg; and (3) diabetes was defined as self-reported diabetes, and/or fasting plasma glucose ≥ 7.0 mmol/L, and/or two-hour plasma glucose ≥ 11.1 mmol/L, and/or HbA1c ≥ 6.5%. Data sources: overweight and obesity: (1) for China, national-average prevalence was from the China Chronic Disease and Risk Factors Surveillance (CCDRFS) program; (2) prevalence in the United States was from the National Health and Nutrition Examination Survey (NHANES); and (3) global prevalence was from the World Health Organization (WHO). Hypertension and diabetes: (1) Chinese national-average prevalence for 2010 and 2013 was from the Report on Chronic Diseases and Risk Factors Surveillance in China (2010 and 2013); for 2018, it was from the Report on Nutrition and Chronic Disease Status of Chinese Residents (2020). (2) For the US, prevalence of hypertension and diabetes was from the National Health and Nutrition Examination Survey (NHANES), 2017–2018. (3) Global prevalence of hypertension was from the World Health Organization (WHO), and prevalence of diabetes among adults (20–79 years) was from the IDF Diabetes Atlas 9th edition, published in 2019.

### 3.3. Awareness, Treatment, and Control Rates of Hypertension and Diabetes, Compared with China’s National Average

Five studies examined the rates of awareness, treatment, and control for hypertension and showed generally low rates [36,37,42,47,59]. Among the five studies, two in Gansu revealed consistently lower awareness, treatment, and control rates for hypertension than China’s national average, three in Lhasa and two influential Buddhist institutes and their surrounding communities showed slightly better, but still low, rates [42,47,59]. The study in Gansu showed the rates monotonically decreased when educational level decreased [36]. A similar monotonic decrease in the rates was also observed in urban, agricultural, and pastoral areas [37]. Two studies suggested local residents had better awareness and treatment rates than monks and nuns [36,42]. We identified two studies of diabetes management among Tibetans, and showed generally lower awareness, treatment, and control rates than China’s national average (Table 1) [38,47].

### 3.4. Risk Factors for Obesity, Hypertension, and Diabetes

We identified 11 studies that examined the environmental, socioeconomic, and lifestyle factors associated with obesity, hypertension, and diabetes among Tibetan adults using multi-variable regression. Nine studies investigated factors for hypertension [31,40,42,46,49,56,59,60,61], while two investigated factors for diabetes and/or pre-diabetes, but none did so for obesity [43,58]. This suggests that more future research is needed on the factors affecting diabetes and obesity risks among Tibetans.

One of the 11 studies included Tibetans from both China and India [43]. Our re-calculated results showed much higher prevalence of overweight/obesity, hypertension, and diabetes among Tibetans living in China than those in India (overweight/obesity, 12.7% vs. 7.2%; hypertension 46.3% vs. 39.1%; diabetes 12.7% vs. 7.2%). More future research is needed to better understand these differences, which can indicate the effects of environmental factors on these health conditions.

Regarding the influencing factors for hypertension, urban residence was consistently shown as an independent risk factor for hypertension (OR 1.46 to 1.99) [31,59]. Living in high-altitude areas was also a consistent risk factor. Depending on cut-off values for “high” altitude and different populations, the ORs ranged from 1.46 to 2.07 [46,56]. Being a monk or a nun was protective for hypertension, compared to being a community resident (OR 0.62) [42]. The association between educational attainment and hypertension was inconsistent, probably due the different reference values for education [49,61]. Regarding lifestyle factors, an inconsistent relationship was shown between hypertension and physical activity [42,61]; two studies identified alcohol drinking as an influencing factor [56,60], while one did not [42]; two identified smoking as risk factor [56,60], while one did not [59]; and one reported that more time spent practicing Buddhist activities decreased hypertension risks [40]. Among dietary factors, vegetarian diet and occasional coffee drinking were associated with decreased ORs for hypertension [31,42]; while excess salt intake and daily fat and oil intake were positively associated with hypertension [31,59].

Regarding the influencing factors for diabetes, farmers and urban dwellers had higher or marginally higher risks for diabetes compared to nomads [43]; neither high altitude nor smoking was associated with the prevalence of diabetes [43,58].

## 4. Discussion

This is the first comprehensive investigation with meta-analysis on obesity and NCD burden among the Tibetan population and highlanders, using findings from 39 published studies and compared with national average estimates for China, the US, and the globe. Our study findings revealed a high burden of obesity and hypertension among the Tibetan population, and the diversified NCD patterns depended on the geographical, environmental, and lifestyle factors. Specifically, pooled-prevalence estimates show almost one-third of Tibetan adults and one-sixth of Tibetan children were overweight or obese, and the estimate of hypertension (31.4%) was higher than recent estimates for China and the globe (27.5% and 22.0%). The low rates of awareness, treatment, and control for hypertension and diabetes further increased the health burden of NCDs. Importantly, high altitude and urban residence were consistent risk factors for hypertension. Only limited studies investigated influencing factors for diabetes, and none have done so for obesity. Further, the finding of the low diabetes prevalence among Tibetans, which did not match the prevalent obesity, raised interesting scientific questions related to public health and clinical management.

However, all these 39 identified studies are based on local study samples and cross-sectional surveys. Neither representative studies of Tibetan population nor longitudinal studies were identified. Future research with such a study design is needed.

### 4.1. Double Burden of Malnutrition (DBM) in Tibetans

Previously, most public health and research efforts regarding Tibetans in China had focused on undernutrition, while not many have focused on NCDs. Available data have shown steady improvements in undernutrition problems in Tibetans, thanks to national and local efforts. Our study findings suggest that future efforts need to address DBM and the increasing NCD burden in Tibetans. The over-nutrition and NCD problems in the Tibetan population in China are likely to become worse in the near future, considering economic development, increase in family income, and shifts in living environment, lifestyle, and occupation factors.

Both DBM across the lifecycle and DBM among children contribute to the prevalent adult obesity. The high pooled prevalence of overweight/obesity (31.6% vs. 39.0% worldwide) among Tibetan adults was probably resultant from suboptimal early life nutrition and the obesogenic environment afterward. Such dynamics are prevalent in indigenous populations [62] and re related to socioeconomic development [63]. A previous study showed Tibetans were at high risk for under-nutrition decades ago [64], which made Tibetans more vulnerable to obesity and other NCDs in lifecycle dynamics. This is shown as undernutrition in early life and overnutrition in adulthood, notably as DBM across the lifecycle [65]. Our previous study revealed a high burden of obesity and NCDs among semi-urbanized Tibetan adults [8,45], which was consistent with the results of this study and to a large international collaborated population study among indigenous populations [4]. Further, this study revealed one-sixth of Tibetan children were overweight or obese. In fact, we found DBM also existed among Tibetan children at the population level [45]. Both conditions—overnutrition and undernutrition—in childhood may predict obesity in adulthood. Public-health-intervention programs for tackling DBM among Tibetan children, such as family-, daycare-, and school-based interventions, are needed in fighting against NCDs and in promoting health equality.

### 4.2. High Hypertension Prevalence among Tibetans

The hypertension burden (high prevalence and low management rates) is high among Tibetans, and this also indicates the burden of the cardiovascular diseases affecting them. Some unique lifestyle and environmental factors, such as excess salt intake and high altitude, may have led to the high hypertension rate among Tibetans [31,46,56]. Excess salt intake is the top-ranking dietary risk factor for health, according to the Global Burden of Disease Project [66]. It has been linked to the common Tibetan practice of drinking butter tea (sūyóu chá), traditionally made from tea leaves, yak butter, water, and salt.

Living in higher altitude areas was a unique risk factor for hypertension. Previous study among inhabitants in the Tibetan areas showed a 2% increase in prevalence of hypertension with every 100 m increase in altitude, which might be in part a result of physiological adaptation to hypobaric hypoxia [67]. The overall health impacts from the adaptation were unknown. Further, altitude-dependent population-specific cut-offs for hypertension are needed. In addition to the interesting scientific question, the altitude threshold for increased hypertension risks and adverse cardiovascular outcomes is also critical for evidence-based health policy and other integrated polices, such as the migration and resettled programs for Tibetans, as we have summarized before [8].

### 4.3. Low Diabetes Prevalence among Tibetans

Future research of diabetes and associated risk factors among Tibetans may provide new insights for the prevention of diabetes. The low pooled-diabetes prevalence among Tibetans (about 40% lower than the national average, 7.5% vs. 11.9%) was consistent with findings from nationally representative surveys [38,51] and may be attributed to multifaceted factors. Current evidence has shown the roles of dietary factors and the diagnostic criteria of diabetes. Our study has identified the metabolic protective role of traditional Tibetan diets, in which tsamba, a whole grain rich in β glucagon functioning in glucose regulation, is an important component [6,68,69].

In addition, the cut-off value of HbA1c for diabetes diagnosis among the Tibetan population, which usually had a higher hemoglobin level due to low oxygen adaptation, should be lower than the normal cut-off [70]. The usage of the identical HbA1c level for diagnosis may lead to an underestimation of diabetes prevalence among Tibetans. Further, studies in other populations, such as Brazilians and Bangladeshis, also showed cut-offs of HbA1c for diabetes diagnosis should be population-specific [71,72]. Tibetan-population-specific HbA1c cut-off, therefore, should be investigated. In addition, HbA1c cut-offs stratified by hemoglobin level needs to be considered.

Another explanation for the low diabetes prevalence may be the relatively low life expectancy of Tibetans. Interestingly, another indigenous highland population in Chile showed a low diabetes prevalence (1.5%) but a high overweight/obesity prevalence (men 39.7%; women 58.0%) in 2001, which was in line with our findings among Tibetans [73]. Underlying genetic, epigenetic, and other mechanisms of low diabetes prevalence in relation to the high-altitude adaption need further investigation.

### 4.4. Diverse NCD Patterns and Associated Risk Factors in Tibetans

The NCD patterns varied some by geographical, environmental, socioeconomic, and lifestyle factors. Of note, the studies included were unevenly distributed by geographic areas. A number of 37 studies were conducted in China, and only 2 relevant studies were in India [7,43]. Of China’s five provincial-level Tibetan areas, over half of the included studies were conducted in the TAR, while only two were in Yunnan. Some of the macro- and meso-level determinants, such as natural and social environments, political systems, and economic development, had a strong influence on obesity and related health outcomes [74], but were largely not studied in the extant literature. This indicates a research gap for further investigation. A unique geographic risk factor for hypertension among Tibetans was high altitude, as discussed above.

Socioeconomic factors may partly explain the differences in prevalence of targeted health outcomes. Lower prevalence of obesity and NCDs exists among Tibetans in India than in China, which was indicated by one included cross-country study in around 2010 [43], which demonstrated the driving force of economic development [63]. This was further supported by the higher pooled prevalence of hypertension and diabetes in urban areas compared to rural areas in this study. The fact that rates of obesity and NCDs were more prevalent among Tibetans in the same Indian community in 2016 lends further credence to this view [7]. In addition to the lagging behind of economic development in the Indian Tibetan community, residents there were the descendants of refugees. The limited resources available may have delayed nutritional and epidemiological transition.

Tibetan Buddhist beliefs and related practices were unique protective factors for obesity and NCD patterns. Our meta-analysis suggested lower pooled-prevalence estimates of hypertension and diabetes among Buddhists in religious institutes compared with urban and rural residents. It also stands to reason that meditative practices may protect Buddhists from hypertension [40,42], and a vegetarian diet may explain the lower prevalence of diabetes [42]. Buddhists also rarely smoke or consume alcohol—lifestyle choices that are linked to NCDs [75].

### 4.5. Hypertension and Diabetes Management

The combination of: (1) low rates of awareness, treatment, and control of NCDs among Tibetans, (2) their high pooled-prevalence estimates of obesity and hypertension; and (3) changes in their living environment and lifestyles suggest a potential future NCD crisis and the challenges that exist for mitigating the impact. Tibetan pastoralists—who live at about 4000 m above sea level—are at the highest risk for hypertension-related adverse outcomes, given their low management rates and high prevalence estimates. This is further complicated by the fact that the large geographic region, sparse population, and limited infrastructure make access to medical care and other social services difficult. In a past study that we conducted within pastoral communities undergoing rapid urbanization, almost three-fourths of the adults that we surveyed had no formal education, and fewer than one in six had completed primary school [6]. This low literacy rate has likely been a barrier to managing NCDs and promoting healthy lifestyles. Given these considerations, Tibetans could potentially benefit from targeted public-health-intervention programs that center on the tenets and practices of Buddhism as well.

### 4.6. Study Strengths and Limitations

The study has some strengths. First, this is the first systematic review and meta-analysis on obesity and related NCD patterns among Tibetans and among highlanders. Second, we reviewed both English and Chinese peer-reviewed publications to gain a comprehensive and culturally informed understanding of the topic. Third, we addressed a clear gap in the literature by considering both natural and social determinants of NCD patterns, such as altitude, livelihood, residence, and religion. In particular, religion’s role in NCD patterns and its potential implications for promoting public health were examined.

This study also has limitations. First, the pooled-prevalence estimates of obesity, hypertension, and diabetes were not standardized. There are considerable variations in study design, study sample selection, and data-analysis methods of the included studies. Second, there was large variability in estimates across reviewed studies. These limited their comparability among the included studies and with the national average of China and other populations. In addition, some of the included studies were of suboptimal quality according to the assessment tool. Nevertheless, we have used the best data available to increase understanding of this important topic.

### 4.7. Policy Implications and Recommendations for Future Research

This study has some important public health and policy implications. First, double-track actions are needed to tackle the ‘double burden’ of increasing the high burden of NCDs and the previously dominant undernutrition and communicable diseases among Tibetans. Second, the higher hypertension prevalence and the lower prevalence of overweight/obesity and diabetes among Tibetans compared with the national average in China, and their poor NCD management showed a critical time window to tackle the NCDs burden among the Tibetan population, in order to avoid worse health and economic consequences in the future. It is necessary to build the capacity of healthcare professionals at the community level, improve the accessibility to healthcare services, and increase the health literacy level among Tibetan residents. Third, the potential health consequences of the increased risks for hypertension in high-altitude areas have important health implications for resettling and relocation programs, which moved many Tibetan pastoralists down to urban areas [8]. Finally, Buddhist institutes and practices can serve as a platform for health promotion and implementation strategies, given the generally low educational level and wide-spread Buddhist beliefs and practices among Tibetans.

Our recommendations for future research include: First, studies using Tibetan-population representative samples and longitudinal studies are needed. This is probably similar for other minority populations in China. A coordinated design and conduction are necessary for such studies. Moreover, translational studies are needed to transfer the identified determinants to evidence-based and culturally sensitive actions. Further, the mechanistic exploration to the low diabetes prevalence among Tibetans and among highlanders may provoke important insights for diabetes prevention and management. Finally, validated cut-offs for hypertension and diabetes diagnosis among Tibetans need to be developed. Possible misclassification of such cases in clinical management is also an interesting and meaningful research question.

## 5. Conclusions

This study revealed the high health burden of obesity and related NCDs among the Tibetan population and the critical time window to tackle obesity and NCDs. There is a need for more rigorous research on this population, including a representative longitudinal study of the Tibetan population and translational and clinical studies to inform tailored interventions and evidence-based, culturally sensitive policy development. Inter- and intra-population health disparity in NCDs should be addressed. In addition, our findings may have some important implications for other indigenous populations in China and for highlanders in other countries, to achieve the Healthy China 2030 national goals and the United Nations’ Sustainable Development Goals.

## Figures and Tables

**Figure 1 ijerph-19-08787-f001:**
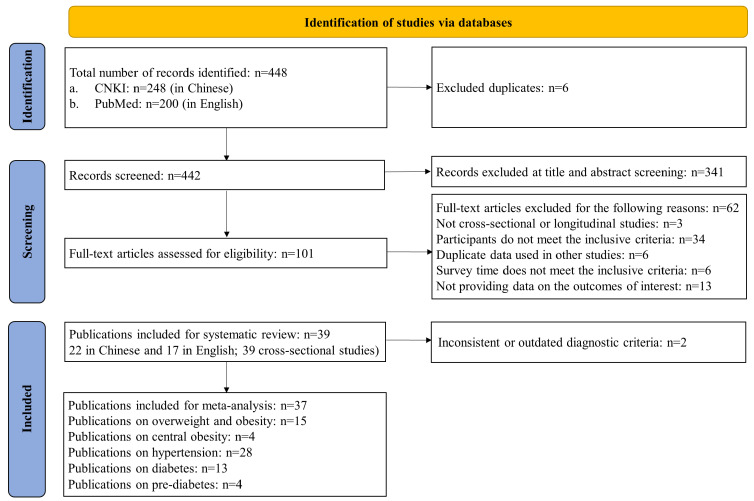
Flowchart of the literature search and study selection according to the PRISMA standard. Abbreviation: PRISMA, Preferred Reporting Items for Systematic Reviews and Meta-Analyses; CNKI: China National Knowledge Infrastructure.

**Figure 2 ijerph-19-08787-f002:**
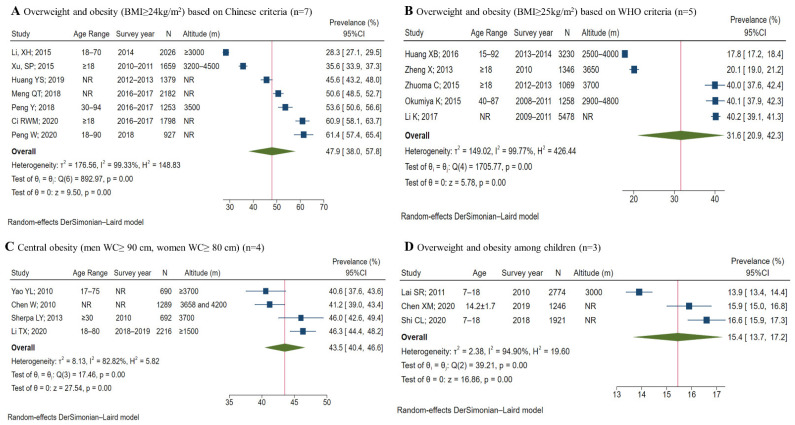
Meta-analysis of pooled prevalence of overweight, obesity, and central obesity among Tibetans [26,27,29,31,32,33,34,35,36,42,43,45,46,47,48,52,55,59,61]. Abbreviations: BMI: body mass index; NR: not reported; M: mean; SD: standard deviation; WHO: World Health Organization; WC: waist circumference. Diagnostic criteria: among adults, overweight and obesity was defined as BMI ≥ 24 kg/m^2^ based on Chinese criteria; overweight and obesity was defined as BMI ≥ 25 kg/m^2^ based on the WHO criteria; central obesity was defined as waist circumference (WC) ≥ 90 cm for men and WC ≥ 80 cm for women. For children, the Chinese BMI criteria for overweight and obesity were used. Prevalence and 95% CI were calculated using random-effects models.

**Figure 3 ijerph-19-08787-f003:**
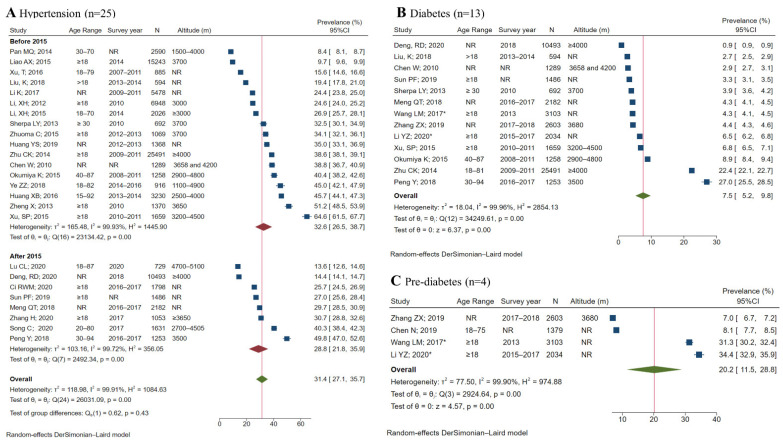
Meta-analysis of pooled prevalence of hypertension and diabetes among Tibetan adults [25,26,29,30,31,32,34,36,37,38,39,40,41,42,43,44,46,47,49,50,51,52,53,56,57,58,59,60,61]. Abbreviations: NR: not reported; M: mean; SD: standard deviation. *: nationally representative survey. Diagnostic criteria: hypertension was defined as systolic blood pressure (SBP) ≥ 140 mmHg, and/or diastolic blood pressure (DBP) ≥ 90 mmHg, and/or a history of hypertension, and/or reported current treatment with antihypertensive medications. Diabetes was defined as self-reported diabetes, and/or fasting plasma glucose (FPG) ≥ 7.0 mmol/L, and/or two-hour plasma glucose ≥ 11.1 mmol/L; pre-diabetes was defined as any participants without diabetes but with an FPG level of 5.6 mmol/L to 6.9 mmol/L, and/or two-hour plasma glucose level of 7.8 mmol/L to 11.0 mmol/L. Prevalence was calculated using the random-effects model. Prevalence and 95% CI were calculated using random-effects models.

**Figure 4 ijerph-19-08787-f004:**
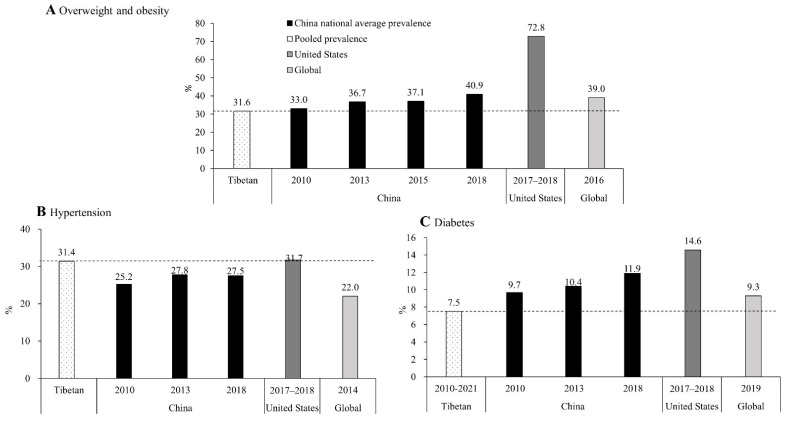
Comparison of combined prevalence of overweight/obesity, hypertension, and diabetes between Tibetan adults (≥18 years) and China, the United States, and the globe.

**Table 1 ijerph-19-08787-t001:** Rates of awareness, treatment, and control of hypertension and diabetes among Tibetan population compared with national average in China.

	Subgroups	Awareness Rate (%)	Treatment Rate (%)	Control Rate (%)	Treatment Control Rate (%)	Interpretation
A Hypertension						
National average	All Male Female Urban Rural	40.9 37.1 45.3 45.4 37.1	32.5 28.6 37.1 38.2 27.7	9.7 9.0 10.5 12.7 7.2	29.8 31.3 28.4 33.1 22.3	Data from China Chronic Disease and Risk Factors Surveillance, 2013.
1. Li X.H., 2015, [37]	Adult residents	16.9 ↓	13.2 ↓	4.5 ↓	NA	Conducted in an impoverished Tibetan Autonomous Prefecture in Gansu. The extreme low rates among monks were due to limited access to civil healthcare services, the belief of predestinated disease by Buddha, and poor healthy literacy.
	Buddhists	9.5 ↓	4.2 ↓	1.6 ↓		
2. Li X.H., 2012, [37]	All Urban Rural	30.4 ↓ 35.7 ↓ 25.2 ↓	20.7 ↓ 25.1 ↓ 16.4 ↓	5.5 ↓ 6.9 ↓ 4.1 ↓	NA	Conducted in an impoverished Tibetan Autonomous Prefecture in Gansu. Low education, health illiteracy, and irregular management may be the reasons for low rates.
3. Meng Q.T., 2018, [42]	Adult residents					Conducted in a Tibetan Autonomous Prefecture in Sichuan. Two temples selected were top-ranking Buddhist institutes. Unlike China’s national data, women usually had worse hypertension-awareness and -management rates than men.
	All Male Female	52.6 ↑ 56.6 ↑ 48.1 ↑	41.3 ↑ 56.6 ↑ 48.1 ↑	3.7 ↓ 4.1 ↓ 3.2 ↓	8.9 ↓ 8.6 ↓ 9.3 ↓	
	Buddhists					
	All Male Female	43.5 ↑ 48.9 ↑ 37.2 ↓	30.1 ↓ 33.3 ↑ 26.4 ↓	6.2 ↓ 8.6 ↓ 3.4 ↓	20.6 ↓ 25.9 ↓ 12.8 ↓	
4. Sherpa L.Y., 2013, [47]	All Male Female	69.4 ↑ 65.3 ↑ 75.2 ↑	59.1 ↑ 54.8 ↑ 62.8 ↑	19.5 ↑ 23.0 ↑ 16.5 ↑	33.0 ↑ 26.3 ↓ 42.1 ↑	Conducted in two counties of Lhasa, the historical and cultural center of Tibetan areas. Study participants had better diabetes management than China’s national average.
5. Zheng X, 2013, [59]	All Male Female	63.5 ↑ 65.9 ↑ 61.6 ↑	24.3 ↓ 25.6 ↓ 23.2 ↓	7.7 ↓ 8.5 ↓ 7.1 ↓	31.8 ↑ 33.3 ↑ 30.4 ↑	Conducted in Lhasa. Most participants were urban residents. The higher educational level among urban residents may explain the better hypertension management compared with other studies.
B Diabetes						
National average	All Male Female	38.6 35.5 42.2	35.6 32.3 39.4	33.0 31.6 34.6	36.3 35.5 37.2	Data from China Chronic Disease and Risk Factors Surveillance, 2013.
1. Sherpa L.Y., 2013, [47]	All Male Female	29.6 ↓ 25.0 ↓ 31.5 ↓	22.2 ↓ 25.0 ↓ 21.0 ↓	7.4 ↓ 0.0 ↓ 10.5 ↓	NA	Conducted in Lhasa. Participants had lower diabetes management than China’s national average.
2. Li Y.Z., 2020, [38]	All	28.3 ↓	43.4 ↑	24.3 ↓	NA	Nationally representative survey, but not Tibetan representative.

Awareness rate was defined as the proportion of individuals with physician-diagnosed disease among all patients. Treatment rate was defined as the proportion of individuals receiving medications, dietary control, and/or increasing physical activity among all patients. Control rate of hypertension was defined as the proportion of individuals with SBP < 140 mmHg and DBP < 90 mmHg among all patients, and treatment control rate of hypertension was defined as the proportion of individuals with SBP < 140 mmHg and DBP < 90 mmHg among patients receiving hypertension treatment. Control rate of diabetes was defined as the proportion of individuals with an HbA1c concentration of <7.0% among all patients with diabetes, and treatment-control rate of diabetes was defined as the proportion of individuals with an HbA1c concentration of <7.0% among patients receiving diabetes treatment. NR: Not available. ↑ indicates higher rate than the national average, and ↓ indicates lower rates.

## Data Availability

Data analyzed in this study were a re-analysis of existing data, which are contained within the article and the Appendix A.

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
