# Peer review of "Prevalence, Management, and Associated Factors of Obesity, Hypertension, and Diabetes in Tibetan Population Compared with China Overall"

_ijerph, 2022, doi:10.3390/ijerph19148787_

Round 1

Reviewer 1 Report

The authors did a good job in this systematic study and meta-analysis on Tibetan population in the context of obesity and non-communicable diseases. Best part of the study is the consideration of social and natural determinants of non-communicable diseases patterns. Despite the heterogeneity  because of the diverse study methods, sample selection, and data analysis methods of the studies included for analysis, the authors did an excellent job with this study which for the first time addressed the gap of Socioeconomic factors in these populations. Specifically, this study addresses role of religion in NCD patterns and its potential implication on public health. Overall, this is an important study which states about extreme burden of obesity and associated comorbidities among Tibetan population, and the critical window of opportunity to deal with obesity and NCDs.

Author Response

The reviewer noted a number of strengths. They felt that the authors did an excellent job with this study which for the first time addressed the gap of socioeconomic factors in Tibetan population. Overall, the reviewer stated that the study is important and the authors did a good job to investigate the extreme burden of obesity and associated comorbidities among Tibetan population. In addition, we conducted high-quality proofreading for the revision. We appreciate the reviewer's comments. 

Reviewer 2 Report

This is a well written and highly relevant article of a systemic research/meta-analysis addressing migration and life style influenced diseases such as obesity, diabetes, hypertension and metabolic syndrome in the Tibetan population inside and outside of Tibet.  

The introduction is well structured and comprehensive. The methods sufficiently explained. The results are well displayed and described.  

The discussion addresses the data adequately and potential bias is being discussed.

As the nature of the work is not political the reasons for migration and relocation of Tibetans out of Tibet have not been addressed. This seems in a medical content partially adequate, but may be a source of bias. The partially fored confrontation of indigenous societies with more developed societies as observed repeatedly in the last two centuries leads to higher rates of obesity, diabetes, hypertension and metabolic syndrome due to the vast access to high calory foods such as sugared drinks etc. Combined with illiteracy and a lack of nutritional knowledge this effect even potentiates. Moreover, the trauma caused by relocation and/or cultural oppression and therefore forced migration may also lead to increased psychological burden and maladapitve behaviors such as alcoholism. Interestingly, the study demonstrates that if a traditional lifestyle is being kept as in the example of the monks low rates of the mentioned health conditions are prevalent. Many interesting topics and issues leading to potential bias in this context remain not answered, but are out of scope of the study.

Besides that from my point of view, everything has been sufficiently addressed.

I understand that Chinese authors my experience problems if health related political issues are made very clear so I don´t expect changes addressing these issues in the current artice.

Author Response

The reviewer noted several merits of the study/manuscript. They found the review well written and highly relevant to mitigating lifestyle influenced diseases such as obesity, diabetes, hypertension, and metabolic syndrome in the Tibetan population inside and outside of Tibet. Furthermore, we appreciate the reviewer’s understanding that this review has limitations due to variations in studies and other potential sources of bias.  We appreciate the reviewer's comments. 

Reviewer 3 Report

This study reports a systematic review of prevalence of obesity, hypertension and diabetes among Tibetans.  Prevalence of obesity was 47.9% among Tibetan adults and 15.4% among children, compared to 51.2% and 19.0% for the overall Chinese population.  Prevalence of hypertension was 31.4% and diabetes 7.5%, compared to 27.5% and 11.9%, respectively, for the overall Chinese population.  Hypertension and diabetes varied by urban/rural residence and residence in Buddhist institutes.  The authors conclude that Tibetans have high burdens of obesity and hypertension, and that representative studies are needed for tailored interventions.

This is an important paper that aims to accurately estimates the prevalence of NCDs in an understudied group, and a group that has lower life expectancy than the overall Chinese population.  

My main comment is that the large variability in estimates across reviewed studies should be emphasized as a caveat, both in the discussion and in the abstract.  For instance, the pooled estimate of obesity using BMI>=25 criteria was 31.6%, but this is based on three estimates of around 40%, one estimate of 17% and one estimate of 20%.  So while obesity in the Tibetan population may well be 31.6%, there is also an argument that it is around 40% (based on three very similar estimates), or possibly that is is around 20% (based on two similar estimates).  There is also substantial heterogeneity for other measures of obesity, hypertension and diabetes. The extent of heterogeneity should be acknowledged.

Other comments are minor.

1. Table 1 is hard to follow and very little of the text is devoted to describing the contents of the table or what it means.  Consider if the table is needed.  If it is, consider redesigning so the contents are more easily understandable.

2. Section 3.1 is labelled "Characteristics of 39 studies included", but it might be better labelled simply, "Studies included", since the section describes the process to identify the 39 studies from those identified and screened, as well as the characteristics of the selected studies.

Author Response

We appreciate the positive feedback as the reviewer noticed that this is an important paper aiming to accurately estimates the prevalence of NCDs in an understudied group, and a group that has lower life expectancy than the overall Chinese population.  

We made the following revisions. 1st, we acknowledged the limitation of large variability in estimates across reviewed studies in both discussion section and in the abstract. Second, we redesigned the table 1. Third, we modified the title of Section 3.1.
